# Optimization of a Green Microwave-Assisted Extraction Method to Obtain Multifunctional Extracts of *Mentha* sp.

**DOI:** 10.3390/foods12102039

**Published:** 2023-05-18

**Authors:** María J. García-Sarrió, María L. Sanz, Jesús Palá-Paúl, Silvia Díaz, Ana C. Soria

**Affiliations:** 1Instituto de Química Orgánica General (CSIC), Juan de la Cierva 3, 28006 Madrid, Spain; mjgarciasarrio@iqog.csic.es (M.J.G.-S.); mlsanz@iqog.csic.es (M.L.S.); 2Departamento de Biodiversidad, Ecología y Evolución, Facultad de Ciencias Biológicas, Universidad Complutense de Madrid, 28040 Madrid, Spain; quibey@bio.ucm.es; 3Departamento de Genética, Fisiología y Microbiología, Facultad de Ciencias Biológicas, Universidad Complutense de Madrid, 28040 Madrid, Spain; silviadi@bio.ucm.es

**Keywords:** *Mentha* sp., microwave assisted extraction (MAE), multifunctional extracts, phenolics, liquid chromatography-mass spectrometry (LC-MS)

## Abstract

A microwave-assisted extraction (MAE) procedure has been optimized to simultaneously provide multifunctional extracts of *Mentha* sp. leaves with improved antioxidant properties and, for the first time, with optimal antimicrobial activity. Among the solvents evaluated, water was selected as the extractant in order to develop a green procedure and also for its improved bioactive properties (higher TPC and *Staphylococcus aureus* inhibition halo). MAE operating conditions were optimized by means of a 3-level factorial experimental design (100 °C, 14.7 min, 1 g of dry leaves/12 mL of water and 1 extraction cycle), and further applied to the extraction of bioactives from 6 different *Mentha* species. A comparative LC-Q MS and LC-QToF MS analysis of these MAE extracts was carried out for the first time in a single study, allowing the characterization of up to 40 phenolics and the quantitation of the most abundant. Antioxidant, antimicrobial (*Staphylococcus aureus*, *Escherichia coli* and *Salmonella typhimurium*) and antifungal (*Candida albicans*) activities of MAE extracts depended on the *Mentha* species considered. In conclusion, the new MAE method developed here is shown as a green and efficient approach to provide multifunctional *Mentha* sp. extracts with an added value as natural food preservatives.

## 1. Introduction

In the last few years, there has been an increased interest in searching for new functional ingredients of natural origin that provide foods with added health benefits [1,2,3]. In addition, despite the existence of a wide variety of food preservation techniques, microbial contamination continues to be an important issue that affects food quality and food security. In this sense, studies on bioprospection of novel extracts with antimicrobial properties and of natural origin (non-contaminant biocides) have attracted great attention [4,5,6,7], and different formulations based on plant extracts or essential oils have been developed as healthier alternatives to synthetic antibacterial/antifungal products, with no effect on food organoleptic properties [8,9].

The genus *Mentha* Linneo belonging to the Lamiaceae family comprises about 19 species of aromatic plants widely distributed through temperate areas of Europe, Asia, Australia, Africa and North America [10]. Besides their use as folk medicine since ancient times and for culinary applications, the large number of biological activities reported for this genus (mainly antimicrobial and antioxidant, but also carminative, expectorant, antitussive, anti-inflammatory, diuretic, etc.) [4,11,12] has promoted the cultivation of mint crops as a new source of nutraceuticals, natural preservatives, etc. that can be applied in the pharmaceutical, cosmetic and food industries, among others.

This demand for biologically active compounds has also encouraged the development of new methodologies to extract *Mentha* sp. bioactives. Whereas the advantages associated with ohmic, microwave or ultrasound-assisted hydrodistillation to obtain essential oils rich in volatile bioactives have been described in a number of studies [13,14,15,16,17], literature regarding the development of procedures based on pressurized liquid extraction (PLE) [18], ultrasound-assisted extraction [19,20] or microwave-assisted extraction (MAE) [21,22,23,24,25] for the enhanced recovery of *Mentha* sp. bioactives is much more limited. However, despite the fact that direct heating of the solvent by microwave irradiation has been shown to improve the speed and efficiency of bioactive extractions and the bioactivity of MAE extracts, previous applications of this advanced technique to *Mentha* sp. have commonly focused on a single *Mentha* species (*Mentha piperita* [21,24,25], *M. pulegium* [22], *M. rotundifolia* [23]), making the comparison of this effect difficult. Moreover, no previous study has addressed the optimization of an MAE method to obtain multifunctional extracts from different *Mentha* species, as the antimicrobial activity of MAE extracts of *Mentha* sp. has never been reported.

Taking into account these antecedents, the aims of the present research were (i) to optimize MAE operating conditions to obtain multifunctional (with antioxidant and antimicrobial activities) extracts of *Mentha* sp. with potential application as food preservatives; (ii) the comprehensive characterization by liquid chromatography coupled to mass spectrometry (LC-MS) of the chemical composition of MAE extracts from different *Mentha* species obtained under previously optimized extraction conditions; and (iii) the evaluation of their bioactivity.

## 2. Materials and Methods

### 2.1. Samples and Standards

Leaves of *Mentha rotundifolia* plants experimentally grown and collected at their flowering stage in 2017 were generously provided by Dr. Burillo of the Centro de Investigación y Tecnología Agroalimentaria de Aragón (Spain). *Mentha* leaf samples of other species (*M. spicata*, *M. suaveolens*, *M. longifolia*, *M. pulegium* and *M. cervina*) were also collected at their flowering stage at different locations in central Spain under botanical surveillance. All samples were air-dried at ambient temperature and in the absence of light, ground in a domestic mill (Moulinex, Barcelona, Spain) and sieved (<500 µm) before MAE extraction.

Caffeic acid, catechin, citric acid, chlorogenic acid, *p*-coumaric acid, diosmetin, epicatechin, *trans*-ferulic acid, gallic acid, gallocatechin, gentisic acid, 4-hydroxybenzoic acid, 3-hydroxy-tyrosol, 3-hydroxy-2,4,5-trimethoxyflavone, isoquercitrin, kaempferol, oxalic acid, quinic acid, syringic acid, apigenin, naringin and vanillic acid were purchased from Sigma-Aldrich (St. Louis, MO, USA). Luteolin 7-*O*-glucoside, salvianolic acid B, scopoletin, hesperetin, diosmin and eriodictyol were acquired from Extrasynthese (Genay Cedex, France), while luteolin was supplied by Sarsyntex (Merignac, France), rosmarinic acid by Cayman Chemical Company (Ann Arbor, MI, USA) and rutin by Acros Organics (Morris Plains, NJ, USA). All standards used were of analytical grade (purity ≥ 95%).

### 2.2. Microwave-Assisted Extraction (MAE)

Microwave-assisted extractions were carried out in a MARS 6 (CEM, Matthews, NC, USA) microwave system equipped with an optical fiber probe for temperature control. The microwave power was set at 900 W. Accurately weighed, powdered dried samples (1 g) of *Mentha rotundifolia* leaves were placed into 100 mL X-Press 1500 vessels (CEM) filled with the selected volume of solvent and subjected to MAE under stirring conditions.

In order to select the best extractant, preliminary experiments were carried out at 75 °C for 17.5 min using 12 mL of different solvents: ultra-pure water (18.2 MΩ·cm) produced in-house using a Milli-Q Advantage A10 system from Millipore (Billerica, MA, USA), ethanol (Scharlau, Barcelona, Spain), methanol (Sigma-Aldrich) and acetone (Carlo Erba, Milan, Italy) of analytical grade. Total phenolic content (TPC) and antimicrobial activity of MAE extracts (determined as described in Section 2.4 and Section 2.6, respectively) were used as criteria for selection of the optimal solvent.

Additional experiments were also performed to optimize the sample/volume ratio (*s*/*V*) by considering 15 mL of water as extractant. The optimal value was chosen to provide the maximum antioxidant activity (DPPH assay, Section 2.5).

MAE was sequentially optimized for 2 independent factors, the extraction temperature (*T*) and time (*t*), by using a 3-level factorial design (Table 1). Experimental ranges for factors evaluated were *T* = 50–100 °C and *t* = 5–30 min. A total of nine experiments were carried out in random order. The quadratic polynomic model proposed was:
*R* = β_0_ + β_1_*T* + β_2_*t* + β_1,1_*T*^2^ + β_2,2_*t*^2^ + β_1,2_*T·t* + ε(1)
where β_0_ is the intercept, β_i_ are the first-order coefficients, β_i,i_ are the quadratic coefficients for ith factors, β_i,j_ are the coefficients for the interaction of factors i and j, and ε is the error. Three response variables were considered in the optimization of the MAE method: the total phenolic content (*R_TPC_*), the antioxidant activity in terms of Trolox equivalents (*R_DPPH_*) and the antimicrobial activity against *Staphylococcus aureus* measured as the growth inhibition halo (*R_AM_*) (Section 2.4, Section 2.5, Section 2.6, respectively). The parameters of the model were estimated by multiple regression using StatGraphics Centurion XVI (v. 16.2.04) software (Statistical Graphics Corporation, Rockville, MD, USA). The experimental conditions to individually maximize *R_TPC_*, *R_DPPH_* and *R_AM_* were obtained from the fitted models. A desirability function (*R_D_*) was also optimized to provide MAE conditions that simultaneously maximize the three above-mentioned responses; this function takes values between 0 (completely undesirable value) and 1 (completely desirable or ideal response). Under optimal experimental conditions, the number of MAE cycles (C1–C3) was also evaluated in terms of the extraction performance for selected phenolics (syringic acid, luteolin-7-*O*-glucoside, rosmarinic acid and salvianolic acid B).

All MAE extracts were immediately cooled down after extraction using an ice-water bath, filtered using Whatman No. 1 filter paper (Sigma-Aldrich) and stored as aqueous solutions in a freezer at −18 °C until analysis. All experiments were carried out in triplicate.

### 2.3. Liquid Chromatography–Mass Spectrometry (LC-MS) Analysis

Two LC-MS instruments (both from Agilent Technologies, Santa Clara, CA, USA) were used in this study to characterize MAE extracts. The first was a LC-UV/MS apparatus consisting of a 1200 Series HPLC system provided with an in-line degasser, a binary pump, an autosampler and a thermostatized column compartment, coupled via an electrospray ionization (ESI) interface working under negative polarity to a single quadrupole MSD 1100 mass spectrometer. The operating parameters of the electrospray source were as follows: spray voltage, 4 kV; drying gas (N_2_, 99.5% purity) temperature, 300 °C; drying gas flow, 12 L min^−1^; nebulizer (N_2_, 99.5% purity) pressure, 276 kPa; and fragmentor voltage, 80–100 V. Quasimolecular ions for target compounds were recorded in the selected ion monitoring (SIM) mode ([M-H]^−^: 197, 447, 359 and 717 for syringic acid, luteolin 7-*O*-glucoside, rosmarinic acid and salvianolic acid B, respectively. Data acquisition and processing were performed using HPChem Station Rev. A.07.01 software.

Initially, the phenolics present in MAE extracts of different *Mentha* species were identified based on the comparison of experimental retention times and mass spectra with data for commercially available standards previously reported in the literature as present in *Mentha* extracts [26,27,28,29]. To confirm these identifications and to characterize unknowns, a liquid chromatography coupled to tandem mass spectrometry LC-MS^2^ system was used. MS^2^ data from literature [27,28,30,31,32] and the Metlin database (Metabolite and Chemical Entity Database, The Scripps Research Institute, San Diego, CA, USA) were also considered.

The instrument employed for these analyses was an Agilent 1200 Series LC system coupled to a 6520 quadrupole-time of flight (QToF) mass spectrometer (Agilent Technologies), using an ESI interface working in the negative-ion mode. The electrospray voltage was set at 4.5 kV, the fragmentor voltage at 150 V and the drying gas temperature at 300 °C. Nitrogen (99.5% purity) was used as the nebulizer (207 kPa) and drying gas (6 L min^−1^), while nitrogen of higher purity (99.999%) was used as the collision gas. Tandem mass spectra were obtained by collision-induced dissociation (CID); collision energies between 10 and 37 eV were applied to the selected precursor ions ([M-H]^−^). Data acquisition and processing were performed using Agilent Mass Hunter Workstation Acquisition Rev. B.02.00 software.

Chromatographic separations were carried out on a reverse-phase Luna C18 analytical column (Phenomenex, Cheshire, UK) (100 × 2.0 mm i.d., 3 µm) operating at a flow rate of 0.4 mL min^−1^. The mobile phase was a binary mixture of solvent A (water with 0.1% acetic acid) and B (acetonitrile with 0.1% acetic acid), according to the following gradient: 0 min: 5% B; 20–25 min: 36% B; 35 min: 90% B; 36–50 min: 5% B. The injection volume was 5 µL, and the column temperature was maintained at 25 °C.

Quantitation of phenolics was performed in triplicate using external standard calibration curves within the 0.01–100 µg·mL^−1^ range. Goodness of fit for these calibration curves and reproducibility of the method were previously assured. Results were expressed as milligrams per gram of dry sample.

### 2.4. Total Phenolic Content (TPC)

The TPC of MAE extracts was determined using the Folin-Ciocalteu (2N) reagent and gallic acid as standard (both from Sigma-Aldrich), according to the method optimized by Singelton et al. [33], with slight modifications. An aliquot (100 μL) of extracts previously diluted (1:50–1:175), 100 μL of MeOH (Sigma-Aldrich) and 100 μL of Folin–Ciocalteu reagent were vortexed in a 1.5 mL Eppendorf. After 5 min, 700 μL of 75 g L^−1^ Na_2_CO_3_ (Panreac, Barcelona, Spain) were added, and the samples were vortexed briefly. The Eppendorfs were then allowed to stand in the dark for 20 min at room temperature. Following this, the samples were centrifuged at 13,000 r.p.m. for 3 min, and their absorbance was read (*n* = 3) at 750 nm. The same procedure was repeated with aqueous solutions of gallic acid in the 10–100 mg L^−1^ concentration range to build up a calibration curve. Results were expressed as gallic acid equivalents (GAE) (mg mL^−1^ or mg·g^−1^ dry sample).

### 2.5. Antioxidant Activity (DPPH Assay)

The antioxidant activity of *Mentha* sp. extracts was measured (*n* = 3) in terms of hydrogen donating or free radical scavenging ability using the stable DPPH radical method [26]. Aliquots (50 µL) of aqueous MAE extracts diluted in methanol in the 1:20–1:70 range were mixed with 45 µL of a 0.001 M methanolic solution of DPPH (Sigma-Aldrich). After 30 min of incubation in darkness at 40 °C, the decrease in absorbance was read at 540 nm.

The inhibition percent of the free radical was estimated using the following equation: *I* (%) = ((*A_control_* − *A_sample_*)/*A_control_*) × 100, where *A_control_* is the absorbance of the control reaction (containing all reagents except the test sample), and *A_sample_* is the absorbance of the test sample. The concentration of extract providing 50% inhibition (*IC*_50_) was estimated from the curve plotting of *I* (%) against the extract concentration. Antioxidant reagent Trolox (Sigma-Aldrich) was used as positive control. Results were also expressed as mg of Trolox equivalents (TE) g^−1^ of dry sample or as mg of TE mL^−1^ of extract.

### 2.6. Antimicrobial Activity

The antibacterial and antifungal activity of MAE extracts of *Mentha* sp. were tested against Gram (+) bacteria (*Staphylococcus aureus*, ATCC 29213), Gram (−) bacteria (*Escherichia coli* (CECT 515) and *Salmonella typhimurium* (CECT 4594)) and against the yeast *Candida albicans* (ATCC 32354). These activities were determined by the agar disk diffusion assay according to Palá-Paúl et al. [34], with slight modifications. Briefly, 100 µL of microbial species from tubes adjusted to the 0.5 MacFarland Standard were inoculated in Petri dishes, using Mueller Hinton (Sigma-Aldrich) as the broth medium for Gram (+) and Gram (−) bacteria and potato dextrose agar (PDA, Sigma-Aldrich) for yeast. Filter paper discs (9 mm in diameter, sterilized at 121 °C and 1 atm) were individually impregnated with 20 µL of extracts previously concentrated 15 times and then placed onto plates. As a negative control, the discs were impregnated with Milli-Q water. After inoculation, all Petri dishes were refrigerated at 4 °C for 15 min to facilitate diffusing the extracts into the medium, and then incubated at 35 °C for 24 h in the case of bacteria, and at 37 °C for 24 h in the case of *C. albicans*. Once the incubation was complete, the diameters of the growth inhibition halos and the bacteriostatic effect (growth rate inhibition) were measured in centimeters (including the diameter of discs). Measurements were carried out in duplicate.

### 2.7. Statistical Analysis

Data were subjected to statistical analysis by using Statistica 7.0 software (StatSoft, Inc., Tulsa, OK, USA). Significance (*p* < 0.05) of differences was evaluated by analysis of variance (ANOVA, Tukey HSD test).

## 3. Results

### 3.1. Optimization of an MAE Method to Provide Multifunctional Extracts of Mentha sp.

#### 3.1.1. Selection of MAE Solvent and *s*/*V* Ratio

As the type of solvent is one of the most relevant factors affecting the extraction of bioactive compounds [35,36], Milli-Q water, ethanol (biosolvent) and renewably sourced extractants, such as methanol and acetone, were evaluated under the same experimental MAE conditions (75 °C and 17.5 min) to obtain *M. rotundifolia* extracts with improved bioactivity in terms of TPC and antimicrobial activity.

As expected for the extraction of polar bioactives, the total phenolic content was shown to increase in the order acetone < ethanol < methanol < water (0.515 < 1.202 < 2.405 < 5.105 mg·GAE mL^−1^). Moreover, the aqueous MAE extract was the only one showing antimicrobial activity against *Staphylococcus aureus* and *Escherichia coli* bacteria and against the yeast *Candida albicans*. Regardless of the solvent considered, no extract was active against *Salmonella typhimurium*. The greatest inhibition halo of bacterial growth provided by this aqueous extract was for *S. aureus* (1.7 ± 0.3 cm) (Appendix A). Therefore, considering that water provided the best results in terms of bioactive properties, and taking into account that the use of non-hazardous and generally recognized as safe solvents is an added value for the development of green and affordable procedures to extract bioactives from natural sources, water was chosen as the optimal extractant for further studies.

In order to select the optimal s/V ratio, results from the extraction of 1 g of *M. rotundifolia* using different water volumes (12 and 15 mL), but identical temperature and time conditions (*T* = 75 °C, *t* = 17.5 min), were compared. As no significant (*p* < 0.05) differences were observed in the antioxidant activity of both experiments (7.1 mg TE mL^−1^), 12 mL of water were selected as optimal to minimize the volume of extractant required.

#### 3.1.2. Optimization of MAE Operating Conditions

The optimization of the extraction temperature and time was subsequently carried out by means of a 3-level factorial experimental design. Table 1 shows the different values obtained for the response variables considered (*R_TPC_*, *R_DPPH_* and *R_AM_*). The antioxidant activity measured in terms of Trolox equivalents (TE) varied between 6.1 and 8.9 mg mL^−1^, and the TPC between 4.5 and 5.8 mg·GAE mL^−1^. In general, the highest values of both responses were obtained at high temperatures (100 °C), while at 50 °C, the extracts presented low TPC concentrations, as well as low antioxidant activity. Regarding the antimicrobial activity (*R_AM_*), extracts obtained at a low temperature (50 °C) showed no activity against *S. aureus*, whereas those subjected to MAE at 75–100 °C gave rise to growth inhibition halos in the 1.1–1.7 cm range (see Appendix A as an example of the *S. aureus* halos obtained under different MAE conditions).

Response surface methodology was further used to calculate the regression coefficients of the models and their statistical significance, as well as the prediction errors (estimation standard error and mean absolute error) (Appendix A). First, the experimental conditions that individually maximized each of the three responses considered in this study were evaluated. High temperatures (100 °C) provided the highest *R_DPPH_* and *R_TPC_*. Regarding the extraction time, *R_TPC_* presented its optimal value (5.91 mg·GAE mL^−1^) for extractions of 30 min, while the maximum response for *R_DPPH_* (9.203 mg TE mL^−1^) was obtained for 5 min extractions. *T* and *T·t* were the most significant (*p* < 0.05) variables for *R_TPC_*, and the equation of the model was *R_TPC_* = 5.024 + 0.0033·*T* − 0.0552·*t* + 0.00074·*T·t* (R^2^ = 91%). As for antioxidant activity, the only significant (*p* < 0.05) variable was *T*, and the equation explaining a percentage high enough of the variability of this response was *R_DPPH_* = 3.446 + 0.0615·*T* + 0.104·*t* − 0.0018·*T·t* (R^2^ = 76%). For antimicrobial activity, high temperatures (93 °C) and intermediate extraction times (19 min) provided the optimal conditions to maximize the growth inhibition halo against *S. aureus* by MAE extracts (*R_AM_* = 1.61 cm). The most significant (*p* < 0.05) factors in this individual optimization were *T* and *T*^2^, and the equation of the model was *R_AM_* = −5.967 + 0.163·*T* − 0.00088·*T*^2^ (R^2^ = 94%). Finally, when a multiple response (*R_D_* = 0.85) that simultaneously maximized *R_DPPH_*, *R_TPC_* and *R_AM_* was considered, 100 °C and 14.7 min were selected as the optimal operating parameters to provide multifunctional extracts of *M. rotundifolia*. Under these conditions, the calculated and experimental responses were very similar, with only *R_DPPH_* showing a higher dispersion (32%) (Table 1).

Under the previously optimized temperature and time conditions, the number of cycles was also evaluated for selected phenolics previously described as major components in *M. rotundifolia* [27,37,38]. Whereas a single cycle was enough to extract 100% of salvianolic acid B and 74–77% of rosmarinic and syringic acids, similar percentages of luteolin-7-*O*-glucoside were extracted in cycles C1 and C2 (44 and 50%, respectively). Percentages lower than 7% of selected phenolics were also extracted in the third extraction cycle. Because of the longer processing time and higher dilution associated with mixing different extraction cycles, a single MAE cycle was considered as a trade-off to recover most of the target bioactives while keeping the required drying step to a minimum for better extract preservation. Therefore, optimal MAE conditions to obtain multifunctional extracts from *M. rotundifolia* leaves were 100 °C, 14.7 min, 1 g of dry leaves/12 mL of water and 1 extraction cycle.

### 3.2. Application of the Optimized MAE Method to Different Mentha Species

#### 3.2.1. LC-MS Characterization

The previously optimized MAE method was applied to obtain multifunctional aqueous extracts from leaves of different *Mentha* species. The compounds present in these extracts were identified by using LC-QToF MS. Appendix A shows the corresponding retention times, quasimolecular ions ([M-H]^−^) and characteristic MS/MS fragments. A total of 40 compounds were identified, 21 of which were unequivocally identified using commercial standards. Whereas some compounds, such as hesperetin, chlorogenic and rosmarinic acids, catechin and gallocatechin isomer, etc., were detected in all the species, most of the compounds were found to be characteristic for some of them. Thus, for example, a higher number of low-molecular-weight compounds (e.g., 4-hydroxybenzoic acid, 3-hydroxy-tyrosol, gentisic acid, etc.) were present in the extracts of *M. longifolia* and *M. cervina*, whereas in the *M. spicata* extract, compounds of higher molecular weight, such as eriocitrin, diosmin, rutin, etc., were detected. Although many of these compounds have already been described in leaf extracts of some *Mentha* species (in *M. spicata* [39], in *M. viridis* and *M. cervina* [28], in *M. rotundifolia* [40], in *M. pulegium* [32]), the identification in the present study of a larger number of compounds in MAE extracts from six different species, extracted and analyzed under identical conditions, makes the comparison of this effect more reliable. Among the phenolic acids described in *Mentha* leaves, it is worth noting the presence of caffeic acid and its derivatives [41], for which a plethora of bioactive properties have been described [42]. Although this acid has been reported in different *Mentha* species, its dimeric (rosmarinic acid) and trimeric (lithospermic acid) forms are the most characteristic phenolic acids of the genus *Mentha* sp. [19]. In this work, caffeic and rosmarinic acids were detected in all the species under study, with the exception of caffeic acid in *M. pulegium*. Different isomers of lithospermic acid were also detected mainly in *M. spicata* and *M. suaveolens*. Chlorogenic acid, obtained by condensation of caffeic and quinic acids, and its isomers were also detected irrespective of the *Mentha* species, while other derivatives, such as caftaric acid or the different isomers of salvianolic acid, were only detected in some species. In addition to caffeic acid derivatives, other acids, such as gentisic acid, cinnamic acid, etc., were also detected in the analyzed extracts.

Regarding flavonoids, flavones are the most characteristic of *Mentha* leaves and, among them, luteolin and its derivatives [41]. Luteolin, luteolin-7-*O*-glucoside and luteolin-7-*O*-rutinoside were detected in almost all the samples analyzed in this study, although with different abundances. However, apigenin, a flavone previously described in some *Mentha* species, such as *M. arvensis* [43] or *M. piperita* [44], or flavanone glycosides, such as naringin, described in extracts of *M. spicata* [26,38,45], were not detected in the MAE extracts obtained in this work. The presence of other flavanones, such as eriocitrin, was observed in *M. spicata* and *M. longifolia*. Flavonols, such as kaempferol, were also detected in *M. pulegium*, *M. longifolia*, *M. rotundifolia* and *M. suaveolens*.

Quantitative analysis (mg·g^−1^ of dry sample) of the main phenolic compounds of MAE extracts from different *Mentha* species was carried out by HPLC-MS in SIM mode (Table 2). Although several studies can be found in the literature comparing the total phenolic content of extracts (generally obtained by solid–liquid extraction or Soxhlet [46,47,48]) from different *Mentha* species, studies related to the individual concentration of these compounds determined by HPLC are more limited [19,26,49]. As expected, the composition of the MAE extracts obtained in this work was different, depending on the *Mentha* species considered. *M. longifolia* and *M. rotundifolia* presented the lowest concentrations of these compounds (4.20 and 6.99 mg·g^−1^ of dry sample, respectively), while *M. suaveolens* had the highest phenolic concentration (14.30 mg·g^−1^ of dry sample).

Among the different *Mentha* species, *M. suaveolens* presented the highest concentration of 4-hydroxybenzoic acid (12.1 mg·g^−1^ of dry sample) and *p*-coumaric acid (0.15 mg·g^−1^ of dry sample) and intermediate levels of rosmarinic acid (1.7 mg·g^−1^ of dry sample), similar to those present in the *M. rotundifolia* extract. These compounds have also been previously detected by Aldogman et al. [50] in *Mentha suaveolens* L. leaves from Saudi Arabia at different levels. Low concentrations of luteolin and caffeic acid, characteristic of *M. suaveolens* ethanolic extracts [26], were also found in this aqueous extract (<0.014 mg·g^−1^ of dry sample).

*M. spicata* presented the highest concentration of rosmarinic acid (4.4 mg·g^−1^ of dry sample), as previously found by Dorman et al. [51] and Rita et al. [52]. Caffeic acid (0.020 mg·g^−1^ of dry sample), chlorogenic acid (0.020 mg·g^−1^ of dry sample) and salvianolic acid B (0.66 mg·g^−1^ of dry sample) were also detected in this sample at concentrations higher than those of other *Mentha* species; these compounds were also reported by Rita et al. [52] in *M. spicata* infusions (concentrations ranging from 1.6 to 7.8 μg·mL^−1^ extract).

As for *M. cervina* extracts, concentrations of 4-hydroxybenzoic acid, syringic acid and rosmarinic acid were found in the range of 3–4 mg·g^−1^ of dry sample. Salvianolic acid A and sagerinic acid were also present at moderate levels in this MAE extract (0.4 mg·g^−1^ dry sample). The presence of these compounds (except for syringic acid) has been previously reported in the literature in infusions and hydroalcoholic extracts of this species [19,28,53].

In contrast to other *Mentha* species, MAE extracts of *M. longifolia* were characterized by low concentrations of rosmarinic acid (0.22 mg·g^−1^ of dry sample) and high concentrations of 4-hydroxybenzoic acid (3.7 mg·g^−1^ of dry sample). On the contrary, Krzyzanowska et al. [49] and Ćavar Zeljković et al. [19] reported higher concentrations of rosmarinic acid (1.9 mg·g^−1^ of dry sample on average) in *M. longifolia* collected in Poland and in Czechia, respectively, and extremely low concentrations of 4-hydroxybenzoic acid (0.83 µg g^−1^ of dry sample). Similarly, Bahadori et al. [54] identified rosmarinic acid (6260 μg·g^−1^ extract) and sinapic acid (7132 μg·g^−1^ extract) as the most abundant compounds in infusions from *M. longifolia* collected in Iran, but 4-hydroxybenzoic acid was not detected. The differences described above may be associated with a number of factors related to the sample analyzed (e.g., where it was harvested, collection period, etc.) or the methodology followed for bioactive extraction (e.g., extractant, technique and operation conditions), among others.

The major phenolic compounds found in the MAE extract of *M. pulegium* were 4-hydroxybenzoic acid and rosmarinic acid (0.45 and 0.15 mg·g^−1^ of dry sample, respectively), followed by quinic acid (0.092 mg·g^−1^ of dry sample), sagerinic acid (0.026 mg·g^−1^ of dry sample) and salvianolic acid A (0.019 mg·g^−1^ of dry sample). These results are in agreement with the composition previously described by Brahmi et al. [38] for ethanolic extracts of this species cultivated in Algeria. However, higher concentrations of 4-hydroxybenzoic acid and rosmarinic acid were found by Ćavar Zeljković et al. [19] in methanolic extracts of this species cultivated in Czechia, while the remaining acids mentioned above were not detected. Proestos et al. [45] reported higher contents of ferulic acid, apigenin and caffeic acid in hydrolyzed *M. pulegium* extracts, compounds not detected or detected at trace levels in the MAE extracts analyzed in this work. This discrepancy is probably due to the fact that these phenolic compounds arise from glycosides that were released during a sample hydrolysis process that was not followed in our study.

Compared to the other *Mentha* species, the phenolic profile of the *M. rotundifolia* MAE extract was characterized by a high content of rosmarinic acid (2.5 mg·g^−1^ of dry sample), as well as high concentrations of syringic acid and salvianolic acid A (1.7 and 1.2 mg·g^−1^ of dry sample, respectively). There is scarce data of individual phenolics present in *M. rotundifolia* extracts in the literature. In a study by Yaiha et al. [55] on the composition of extracts of *M. rotundifolia* leaves from different geographical areas of Tunisia, rosmarinic acid was also found as one of the most abundant phenolics (ranging between 6.53 and 116.15 mg·g^−1^ extract). However, salvianolic acid A and syringic acid were not detected. These authors also reported a significant qualitative and quantitative phenolic variability between the studied samples, probably due to the wide diversity of ecological factors characterizing the sites where samples had been collected.

#### 3.2.2. Total Phenolic Content and Antioxidant Activity

The total phenolic content and antioxidant activity (determined by the DPPH method) of MAE extracts from the different *Mentha* species under study are shown in Table 3. *M. spicata* and *M. rotundifolia* extracts showed significantly higher TPC values (48 and 48.6 mg·GAE·g^−1^ of dry sample, respectively) than the other *Mentha* species, with *M. pulegium* extract showing the lowest value (18.7 mg·GAE·g^−1^ of dry sample). A wide variability in TPC has also been described in previous studies for extracts of different *Mentha* species. Thus, in a recent study by Ćavar Zeljković et al. [19], methanolic extracts of *M. suaveolens* presented the highest TPC (58.93 mg·GAE·g^−1^), while *M. cervina* extracts showed the lowest (14.81 mg·GAE·g^−1^).

As reported in the literature [47], essential oils and extracts of some of the major genera belonging to the Lamiaceae family (e.g., *Mentha*, *Salvia*, *Sideritis*, etc.) have been described to possess antioxidant activity. However, whereas the number of papers dealing with the TPC and antioxidant activity of *Mentha* sp. essential oils and extracts (usually obtained by conventional extraction) is high [11,26,56], only two references have reported to date the TPC and antioxidant activity of MAE extracts, and they only evaluate a single *Mentha* species (*Mentha piperita*) [24,25].

In general, the aqueous MAE extracts obtained in this work showed higher TPC values (18.7–48.6 mg·GAE·g^−1^ of dry sample) than alcoholic (ethanol and methanol) extracts obtained by conventional extraction from different *Mentha* species collected in Algeria (*M. rotundifolia*, *M. spicata* and *M. pulegium*: 4.6–12.0 mg·GAE·g^−1^ of dry sample) [38] and in Greece (*Mentha viridis* and *Mentha pulegium*: 16.5 and 8.4 mg·GAE·g^−1^ dry sample, respectively) [45], and similar values to those obtained from *M. piperita* by MAE using methylene chloride as solvent (40.49 mg·GAE·g^−1^) [25]. However, the results listed in Table 3 were slightly lower than those of MAE extracts of *M. piperita* from Serbia, obtained using 40–80% ethanol as extractant (64–113 mg·GAE·g^−1^ of dry sample) [24], and those reported for 70% ethanolic extracts of different *Mentha* species from Romania (51–98 mg·GAE·g^−1^ of dry sample). Similar to the results described here, this last study found the highest concentrations of total phenolic compounds in *M. rotundifolia*, followed by *M. spicata* and *M. suaveolens* [26].

The highest ability for DPPH free radical scavenging was found in *M. spicata* (120 mg·TE·g^−1^ dry sample), followed by *M. suaveolens* (69 mg·TE·g^−1^ dry sample); no significant differences were found in the remaining samples evaluated (49–57 mg·TE·g^−1^ of dry sample). These results are in agreement with those of Pavlic et al. [24], who reported antioxidant activities of MAE ethanolic peppermint extracts in the same range (67–122 mg·TE·g^−1^ of dry sample), and are higher than those obtained from this *Mentha* specie using methylene chloride as MAE extractant (19.79–69.15 mg·TE·g^−1^) [25].

The *IC*_50_ values for aqueous extracts obtained by infusion and by SLE from *M. cervina* [28], and by SLE from *M. spicata* [52], were higher (210–810 μg·mL^−1^) than those of the aqueous MAE extracts obtained in this work from the same mint species (Table 3). Compared to well-recognized antioxidants, the *IC*_50_ values determined for MAE extracts optimized in this paper were 4–13 times higher than Trolox (*IC*_50_ = 27 μg·mL^−1^). A similar antioxidant activity (*IC*_50_ values in the range 4–16 times higher than Trolox) has been previously reported for ethanolic extracts obtained at room temperature, for 24 h, from leaves of *M. spicata*, *M. pulegium* and *M. rotundifolia* collected in Algeria [38]. Slightly higher *IC*_50_ values (10–20 times higher than Trolox, in the range 105–285 μg·mL^−1^) have also been previously reported by Moldovan et al. [26] for 70% ethanolic extracts of different Romanian *Mentha* species. In agreement with previous references [11,45,52,57], the different *Mentha* species, harvesting locations and efficiency for bioactive extraction of all these methodologies could be responsible for the variations observed regarding both the TPC content and antioxidant activity of *Mentha* extracts.

Although some similarities between the TPC and DPPH results were observed (e.g., data for *M. spicata*), no significant correlations were found between the results of these two assays. Therefore, it is likely that the type and concentration of phenolic compounds, as well as the presence of other compounds with antioxidant activity in MAE extracts, could be responsible for this lack of correlation. Similar results have been previously described by other authors in studies on the antioxidant activity of different types of samples [45,47].

#### 3.2.3. Antimicrobial Activity

Table 3 shows the growth inhibition halos of Gram (+) (*Staphylococcus aureus*) and Gram (−) (*Escherichia coli* and *Salmonella typhimurium*) bacteria and of the yeast *Candida albicans* in the presence of the MAE extracts of the different *Mentha* species. Both the bactericidal and bacteriostatic effect of each extract were considered for the different strains. It is known that the antimicrobial and antifungal activities depend on the *Mentha* species considered [26,45]. Although it has been described that Gram (−) bacteria are more resistant than Gram (+) to the antibacterial properties of essential oils and plant extracts [11,12], the MAE extracts obtained in this work under optimal operating conditions did not show a selective activity based on the differences between the cell walls of the different bacterial microorganisms tested. Thus, while the MAE extracts of all the *Mentha* species evaluated, except for *M. rotundifolia*, presented moderate bactericidal activity against *E. coli* (with a mean bacterial growth inhibition halo of 1.0 cm), the *M. cervina*, *M. suaveolens* and *M. pulegium* extracts were the only ones that showed an additional bacteriostatic effect (2.4–2.6 cm halo).

*Mentha* species other than *M. suaveolens* were also active against *S. aureus* (growth inhibition halo in the 1.0–1.7 cm range). Similarly to previous references [26], *M. rotundifolia* and *M. spicata* extracts were found to be the most active in inhibiting the growth of this microorganism. In addition, *M. pulegium* and *M. cervina* were the only samples showing a bacteriostatic effect against *S. aureus*, with no significant differences in the activity determined for both extracts (3.2–3.3 cm halo). Regarding *S. typhimurium*, all *Mentha* sp. extracts assayed failed to inhibit the growth of this Gram (−) bacteria. *M. cervina* was the only species that was active against *C. albicans* (inhibition halo of 1.7 cm).

Although to a lesser extent than essential oils reported in the literature for several *Mentha* species [11,12], MAE extracts were found to be active against different bacterial and fungal strains, with *M. pulegium* and *M. cervina* showing the widest inhibitory spectrum. As for different extraction techniques, the MAE aqueous extracts obtained in this work compared favorably with respect to Soxhlet methanolic extracts of *M. longifolia* [11] and with aqueous extracts obtained by accelerated solvent extraction from *M. longifolia* and *M. pulegium* [47], and were of the same magnitude as 70% ethanolic extracts obtained by conventional extraction from *M. spicata*, *M. suaveolens* and *M. rotundifolia* [26].

## 4. Conclusions

In conclusion, a fast, efficient and environmentally friendly MAE method (using water as the solvent) has been fully optimized, considering a multiple response based on antioxidant capacity and, for the first time, antimicrobial activity, to obtain multifunctional extracts from six different *Mentha* species (*M. spicata*, *M. pullegium*, *M. suaveolens*, *M. longifolia*, *M. cervina* and *M. rotundifolia*). In agreement with the variable phenolic composition determined by LC-Q MS and LC-QToF MS, the bioactivity of MAE extracts was found to be highly dependent on the *Mentha* sp. considered. This contribution, in which MAE extracts from six different *Mentha* species were obtained under identical operating conditions, represents an advantage over previous references, in which the variety of extraction techniques, bioactivity assays, etc. applied to a single or very limited number of *Mentha* species did not allow the proper comparison of this effect. Moreover, this study evaluates the antimicrobial (antibacterial and antifungal) activity of *Mentha* MAE extracts for the first time, thereby extending the multifunctionality of these antioxidant-rich extracts and the number of potential applications (e.g., natural preservatives) in the food industry, among others.

## Figures and Tables

**Table 1 foods-12-02039-t001:** Results of the experimental design for the optimization of MAE conditions to extract bioactive compounds from *M. rotundifolia* leaves.

Experiment	*T* (°C)	*t* (min)	*R_TPC_* (mg·GAE mL^−1^)	*R_DPPH_* (mg TE mL^−1^)	*R_AM_* (cm)
1	50	17.5	4.83 (0.09) *	6.9 (0.1)	-
2	50	5	5.12 (0.06)	6.1 (0.3)	-
3	50	30	4.5 (0.4)	6.8 (0.5)	-
4	75	5	5.45 (0.06)	8.9 (0.3)	1.2 (0.1)
5	75	30	5.48 (0.04)	7.3 (0.4)	1.1 (0.1)
6	100	30	5.80 (0.08)	7.1 (0.2)	1.4 (0.3)
7	75	17.5	5.2 (0.1)	7.1 (0.3)	1.7 (0.3)
8	100	17.5	5.58 (0.05)	8.4 (0.1)	1.7 (0.3)
9	100	5	5.5 (0.1)	8.7 (0.7)	1.6 (0.2)
*R_D_*	100	14.7			
Calculated responses			5.45	9.2	1.57
Experimental responses			5.26	6.18	1.70

* Mean values and standard deviation in parentheses (*n* = 3).

**Table 2 foods-12-02039-t002:** Concentration (mg·g^−1^ of dry sample) of the main phenolic compounds of MAE extracts obtained for different *Mentha* species.

Compound	*M. spicata*	*M. pulegium*	*M. cervina*	*M. longifolia*	*M. rotundifolia*	*M. suaveolens*
4-Hydroxybenzoic acid	- ^a^	0.45 (0.09) ^a^*	4.36 (0.07) ^b^	3.7 (0.4) ^b^	0.69 (0.02) ^a^	12.1 (0.9) ^c^
3-Hydroxy-tyrosol	- ^a^	- ^a^	0.033 (0.004) ^c^	0.0243 (0.0004) ^b^	- ^a^	0.033 (0.005) ^c^
*p*-Coumaric acid	tr ^a^	tr ^a^	tr ^a^	0.030 (0.001) ^a^	- ^a^	0.15 (0.06) ^b^
Vanillic acid	- ^a^	0.0122 (0.0003) ^a,b^	0.03 (0.01) ^b,c^	0.05 (0.04) ^c^	- ^a^	- ^a^
Caffeic acid	0.020 (0.001) ^b^	- ^a^	tr ^a^	0.010 (0.004) ^b^	0.043 (0.008) ^c^	0.014 (0.001) ^b^
Quinic acid	- ^a^	0.092 (0.009) ^b^	0.08 (0.01) ^b^	- ^a^	0.15 (0.02) ^c^	- ^a^
*trans*-Ferulic acid	tr	tr	tr	-	-	-
Syringic acid	0.5 (0.1) ^a^	- ^a^	4.1 (0.9) ^c^	0.14 (0.05) ^a^	1.7 (0.7) ^b^	- ^a^
Luteolin	0.0005 (0.00002) ^c^	0.0002 (0.0001) ^a,b^	- ^a^	0.0003 (0.0002) ^b^	tr ^a^	0.00016 (0.00005) ^a,b^
Chlorogenic acid	0.020 (0.005) ^d^	0.009 (0.001) ^c^	0.016 (0.005) ^d^	0.006 (0.001) ^b,c^	0.0033 (0.0007) ^a,b^	tr ^a^
Rosmarinic acid	4.4 (0.8) ^d^	0.15 (0.01) ^a^	3.2 (0.9) ^c^	0.22 (0.01) ^a^	2.5 (0.1) ^b,c^	1.7 (0.5) ^b^
Luteolin-7-*O*-glucoside	0.004 (0.001) ^b^	0.00003 (0.00001) ^a^	- ^a^	- ^a^	0.002 (0.002) ^b^	- ^a^
Salvianolic acid A	- ^a^	0.019 (0.001) ^a^	0.4 (0.1) ^b^	- ^a^	1.2 (0.2) ^c^	- ^a^
Salvianolic acid B	0.66 (0.05) ^c^	0.0126 (0.0002) ^a^	tr ^a^	0.0204 (0.0002) ^a^	0.5 (0.1) ^b^	- ^a^
Sagerinic acid	- ^a^	0.026 (0.001) ^a^	0.4 (0.1) ^c^	- ^a^	0.2 (0.1) ^a,b^	0.3 (0.1) ^b,c^

* Mean values and standard deviation in parentheses (*n* = 3), -: non detected (below the *LOD* in the range 0.0024–0.006 μg·g^−1^ depending on the compound), tr: traces (below the *LOQ* in the range 0.0084 - 0.0192 μg·g^−1^ depending on the compound). ^a–d^ Different letters within the same row indicate significant differences (*p* < 0.05) among extracts of the different *Mentha* species.

**Table 3 foods-12-02039-t003:** Total phenolic content (TPC), antioxidant (DPPH method) and antimicrobial activities of MAE extracts of different *Mentha* species.

			*M. spicata*	*M. suaveolens*	*M. longifolium*	*M. pulegium*	*M. cervina*	*M. rotundifolia*
Growth inhibition halo (cm)	Bactericidal effect	*E. coli*	1.00 (0.05) *^,a^	1.00 (0.05) ^a^	1.00 (0.05) ^a^	1.00 (0.05) ^a^	1.00 (0.05) ^a^	-
		*S. typhimurium*	-	-	-	-	-	-
		*S. aureus*	1.35 (0.07) ^b^	-	1.00 (0.05) ^a^	1.00 (0.05) ^a^	1.00 (0.05) ^a^	1.7 (0.3) ^b^
	Bacteriostatic effect	*E. coli*	-	2.4 (0.2) ^a^	-	2.5 (0.3) ^a^	2.6 (0.1) ^a^	-
		*S. aureus*	-	-	-	3.2 (0.2) ^a^	3.3 (0.3) ^a^	-
	*C. albicans*		-	-	-	-	1.7 (0.3)	-
TPC (mg·GAE·g^−1^ of dry sample) **			48 (1) ^c^	34.0 (0.2) ^b^	26.2 (0.7) ^b^	18.7 (0.1) ^a^	30.72 (0.07) ^b^	48.6 (0.6) ^c^
DPPH (mg·TE·g^−1^ of dry sample) ***			120 (9) ^c^	69 (5) ^b^	51 (4) ^a^	55 (2) ^a^	49 (7) ^a^	57 (2) ^a^
*IC*_50_ (μg·mL^−1^)			99 (6) ^a^	118 (7) ^a^	247 (9) ^c^	359 (6) ^d^	150 (3) ^b^	168 (5) ^b^

* Standard deviation in parentheses (*n* = 3); ** GAE: gallic acid equivalents; *** TE: Trolox equivalents. ^a–d^ Different letters within the same row indicate significant (*p* < 0.05) differences among extracts of the different *Mentha* species.

## Data Availability

Data is contained within the article or Appendix A.

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
