# Peer review of "Optimization of a Green Microwave-Assisted Extraction Method to Obtain Multifunctional Extracts of Mentha sp."

_foods, 2023, doi:10.3390/foods12102039_

Round 1

Reviewer 1 Report

1. the lack of novelty is visible in the paper (the authors were asked to better explain the basic strengths and weaknesses of their approach, and to put a new insight for better explain the novelty of the discussion), 2, the deficiency of the experimental results were especially focussed in my review! (if the authors were unable to present more better results and the more adequate experimental design, it must be mentioned in the paper, perhaps that issue will be resolved in their future work, but it must be explained)! This issue was also mentioned in my review. The results of the numerical study must be compared to the experimental results! A beautiful graph doesn't mean that the mathematical model was accurate? 3. the verification of the model is missing! The reader would not believe the author's statement of the model's accuracy if the model was not verified! Where is the residual analysis? How the reader would trust the authors when these issues were not even mentioned in the manuscript? All those aspects are mentioned in my review? I know that my review is not quit possitive to the authors, but I think the authors should understand that the reviewer`s comments contribute to better quality of the paper that was submitted, and that our work is not targeted against their behoofs, but rather for their own good.

Reviewer 2 Report

The manuscript by Garcia-Sarrio et al al. reports the study of the microwave assisted extraction (MAE) to obtain extracts from Mentha Sp. leaves. The experimental methodology included the use of design of experiment (Box-Benkel design) with the objective of optimizing the MAE operational parameters (time and temperature) for the maximum of total phenol content, antioxidant activity and antimicrobial activity of the extracts. Moreover, using the optimized extraction conditions MAE was applied to six different Mentha species and the obtained extracts were characterized in terms of its phenolic compounds composition.

The research work concerns the study of microwave assisted extraction process, which is in accordance with the Green Chemistry Principles, i.e. the use of more sustainable and green processes. Also, the authors selected a green solvent, water, which highlights the sustainability of the process.  In this regard, the present study addresses a topic of interest.

This is in general a very good paper. The findings are interesting and contribute to the literature. The research methodology is sound.
I recommend the publication of the manuscript with revision of the English. Moreover, listed bellow are some questions regarding the presented work, which the authors could clarify:

Section 2 – Materials and methods: Did the authors characterize the Mentha leaves with a standard method? (. in order to assess the composition in TPC of these leaves?). This procedure could enable to assess the efficiency of the MAE method towards obtaining these compounds from the Mentha leaves.

If so, please add this information to the text.

Section 2.2 MAE, last paragraph (lines 126-128) – Was the water removed from the MAE extract? Or was the extract storage as an aqueous solution. Please clarify this in the text.

Section 2.5 – The aliquots of MAE extracts were used as obtained from the extraction procedure (e.g. , aqueous extract) or was the water evaporated and then the extracts re-dissolved in methanol? Please add this details to the procedure.

Round 2

Reviewer 1 Report

The Authors managed to improve the qyality of the Manuscript according to the Reviewers' comments. I suggest the Editor to accept the Manuscript in the presnted form, for possible publication in the Foods.